# Improved Differential Privacy for SGD via Optimal Private Linear Operators on Adaptive Streams

**Sergey Denisov**
University of Wisconsin-Madison
denissov@wisc.edu

**H. Brendan McMahan**
Google Research
mcmahan@google.com

**Keith Rush**
Google Research
krush@google.com

**Adam Smith**
Boston University
ads22@bu.edu

**Abhradeep Thakurta**
Google Research
athakurta@google.com

## Abstract

Motivated by recent applications requiring differential privacy over adaptive streams, we investigate optimal instantiations of the matrix mechanism [1] in this setting. We prove fundamental theoretical results on the applicability of matrix factorizations to adaptive streams, and provide a parameter-free fixed-point algorithm for computing optimal factorizations. We instantiate this framework with respect to concrete matrices which arise naturally in machine learning, and train user-level differentially private models with the resulting optimal mechanisms, yielding significant improvements in a notable problem in federated learning with user-level differential privacy.

## 1 Introduction and background

An important setting for private data analysis is that of streaming inputs and outputs—often dubbed *continual release*. Hiding individual information is especially challenging in such settings since the arrival of one person's data may affect all future outputs of the system. A significant line of work formalizes *differential privacy* (DP, [2]) under continual release and builds algorithms that meet the resulting definition (e.g. [3–8]). One prominent application of private, continual-release algorithms is to adapt iterative optimization algorithms such as SGD so that their outputs satisfy DP [7, 6, 9].

The problem of privately computing *cumulative sums* plays a key role in both theory and applications. Given a set of input vectors (e.g., gradients) $\mathbf{g}_1, \ldots, \mathbf{g}_n$ with $\mathbf{g}_i \in \mathbb{R}^d$, the task is to approximate the sequence of prefix sums $(\mathbf{g}_1, \mathbf{g}_1 + \mathbf{g}_2, \ldots, \mathbf{g}_1 + \cdots + \mathbf{g}_n)$ while satisfying DP. Solutions to this task form the core building block in DP algorithms for online PCA [10], online marginal estimation [4, 3, 11], online top-k selection [12], and training ML models [13, 6, 14], among others. For example, a common approach to private optimization is to add noise to the gradient estimates in SGD [15–17]. Kairouz et al. [6] make the observation that the key DP primitive in such contexts is not the independent estimation of individual gradients, but rather the accurate estimation of cumulative sums of gradients. Lowering the error of the DP algorithm's approximation to the cumulative sum translates directly to improved optimization.[1]

---

[1] SGD with constant learning rate $\eta$ serves as an intuitive illustration: performing SGD on parameters $\theta$ starting from $\mathbf{0}$, the $t$-th iterate is simply $\theta_t = -\eta \sum_{i=1}^{t} \mathbf{g}_i$, where $\mathbf{g}_i$ is the gradient computed on step $i$. That is, the learned model parameters $\theta_t$ are exactly a scaled version of the cumulative sum of gradients so far. It is the total error in these cumulative sums that matters most, not the error in the private estimates of each individual $\mathbf{g}_i$. See Theorem 5.1 of [6] for a formal statement.

36th Conference on Neural Information Processing Systems (NeurIPS 2022).

The structure of continual-release algorithms imposes two major constraints: first, the algorithm must be computable online—that is, we must produce the output at a given time using only prior inputs—and second, privacy analysis must account for *adaptively defined inputs*—that is, the guarantee should hold even against an adversary that selects inputs based on all previous outputs of the system. In learning applications, the privacy analysis must be adaptive even when the stream or raw training examples is fixed in advance, because the points at which we compute gradients depend adaptively on the output of the mechanism so far [14].

In this paper, we revisit the design of continual-release algorithms for cumulative sums and related problems. We give new tools for analyzing privacy in the adaptive setting, new methods to design optimal (within a class) algorithms for summation-style problems, and applications to central problems in private machine learning. Although we focus on learning as the primary application of our algorithmic and analytic tools, our techniques apply to a wide range of private computations over streaming data such as online monitoring [18, 19], tracking distributional changes over data streams [20], and detection of emerging trends [21].

**Prior Approaches**    Prior approaches to DP approximation of cumulative sums fall broadly into two categories. First, in streaming settings, existing work is generally based on the *binary-tree estimator* [4, 3]. This estimator embeds the values to be summed as leaf nodes in a complete binary tree $\mathcal{T}$, with internal nodes representing the sum of all leaves below them. The mechanism views the *entire tree* as the object to be privately released (ensuring privacy by adding independent noise to each node). An important refinement of Honaker [22] leverages the multiple independent noisy observations of correlated values to produce lower-variance estimates of the prefix sums. Kairouz et al. [6] apply Honaker's online variant, dubbed `Honaker Online`, to the *follow-the-regularized-leader* approach to optimization [23–25]. The tree structure of these mechanisms allows for privacy analysis in the adaptive setting, as observed by [14] and formalized by [8].

The second, more general approach to cumulative sums has previously only been applied in *offline* settings, in which the input is received and outputs are produced as a single batch. The idea is to view cumulative sums as a special case of linear query release (since each output is a pre-specified linear combination of the inputs). In the offline setting, the tree-based approaches can be viewed as instantiations of this widely-studied *factorization* framework (as in, for example, [26, 1, 27–29]). To introduce the general matrix factorization approach, consider the task of computing a private estimate of a linear mapping $\mathbf{G} \mapsto \mathbf{AG}$ defined by matrix $\mathbf{A} \in \mathbb{R}^{n \times n}$ (assumed to be full-rank throughout this work). Given any factorization $\mathbf{A} = \mathbf{BC}$, a DP estimate of $\mathbf{AG}$ can be computed as

$$\widehat{\mathbf{AG}} = \mathbf{B}\left(\mathbf{CG} + \mathbf{Z}\right) \tag{1}$$

where $\mathbf{Z}$ represents a sample from a noise distribution $\mathcal{D}$. Choosing $\mathcal{D}$ in a way that appropriately depends on the sensitivity of the map $\mathbf{G} \mapsto \mathbf{CG}$, one can prove privacy of the noised vector $\mathbf{CG} + \mathbf{Z}$, and hence of the mapping $\mathbf{G} \mapsto \widehat{\mathbf{AG}}$. For example, in the case of cumulative sums, the matrix $\mathbf{A}$ is the lower-triangular matrix $\mathbf{S}$ with 1's on and below the diagonal; the binary tree mechanisms correspond to a matrix $\mathbf{C}_{\mathcal{T}}$ with one row per tree node (see Appendix A and Appendix C respectively). A typical choice for the noise $\mathbf{Z}$ is to select it from a spherical Gaussian distribution.

A focus of existing work (e.g. [1, 27–29]) is to choose the factorization $\mathbf{BC}$ to optimize some measure of overall accuracy (such as total mean squared error) subject to a privacy constraint. However (with the exception of the independent, parallel work of Fichtenberger et al. [30][2]), streaming constraints and adaptive privacy were not explicitly considered. In the streaming setting, one naturally requires the $i^{th}$ element (row) of $\mathbf{AG}$ to be computable using only the first $i$ elements (rows) of $\mathbf{G}$. This corresponds to requiring that the linear operator of interest $\mathbf{A}$ has a lower-triangular structure when represented as a matrix, a requirement that is met by all the matrices under consideration in this work. Distributing the $\mathbf{B}$ in Eq. (1), we see $\widehat{\mathbf{AG}} = \mathbf{AG} + \mathbf{BZ}$. As long as $\mathbf{AG}$ is lower triangular, then, *any* matrix mechanism can be implemented by an online algorithm, via the distribution induced by $\mathbf{BZ}$. However, this observation does not address a key problem: under what conditions is the resulting mechanism *adaptively* private?

---

[2]Fichtenberger et al. [30] strive to a get an analytical optimal leading multiplicative constant for the additive error achievable for the problem of continual observation under DP. We on the other hand focus on computationally estimating the optimal matrix factorization mechanism for the problem under DP. We leave the empirical comparison to the explicit construction in Fichtenberger et al. for future work.

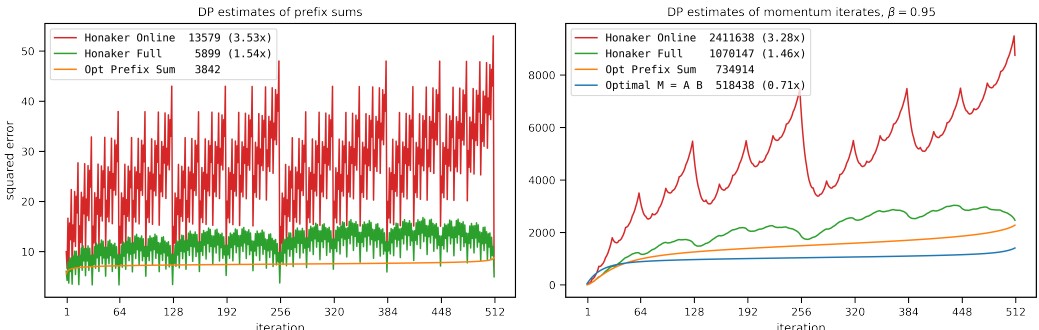

Figure 1: Left: Per-iteration squared error of three mechanisms for DP online prefix sums for $n = 512$. `Honaker Online` and `Honaker Full` correspond to DP binary tree aggregation with the streaming and full Honaker estimators (respectively), while `Opt Prefix Sum` corresponds to the optimal matrix factorization. The tree-based mechanisms suffer from variability in the error due to the binary tree structure. Right: Squared error in the DP estimates of the iterates of momentum SGD for four mechanisms. Momentum is treated as post-processing of cumulative sums for the first 3 mechanisms, while `Optimal M = B C` uses the optimal factorization of the momentum matrix (see Section 4).

The applications to optimization raise their own set of critical questions: How can we efficiently compute such factorizations? Which linear operators are the appropriate ones to factorize in the case of SGD? Finally, can these factorizations actually produce improved privacy/accuracy tradeoffs for real-world machine learning tasks?

**Contributions** We provide a deeper understanding of the matrix mechanism in the adaptive streaming setting. We show that if $\mathbf{Z}$ in Eq. (1) is drawn from a Gaussian distribution of appropriately computed variance, then the resulting mechanism is differentially private in the adaptive streaming setting, *independent of the structure of* $\mathbf{B}$ *and* $\mathbf{C}$. We make an explicit connection here with the easier-to-see privacy under adaptive streams of lower-triangular factorizations. Furthermore, we show that this property is specific to the Gaussian mechanism, and does not extend to arbitrary noise distributions.

For natural notions of error and adjacency specified in Section 3, we present a fast and parameter-free fixed-point algorithm for computing optimal factorizations and prove a local convergence guarantee for this algorithm, leveraging representations of optimal factorizations which are to our knowledge novel in the literature on the matrix mechanism. The optimal computed factorizations show a significant improvement over existing state-of-the-art private streaming prefix sum methods [22], additionally removing the artifacts of the binary tree data structure (see Fig. 1). Furthermore, our fixed-point algorithm can be two orders of magnitude more computationally efficient than existing optimization methods for the matrix mechanism [27] (though we did not explicitly compare to McKenna et al. [28]), and provides a direct bound on the duality gap which allows precise stopping criteria.

Going beyond prefix sums (which correspond to constant learning rate SGD as noted above), we construct matrix mechanisms that directly encode more sophisticated optimization algorithms as linear operators on gradients: in particular, arbitrary combinations of (data independent) learning rate schedules and momentum.

We compute optimal factorizations of these general matrices via fixed-point iterations, and use them to train user-level differentially private language models on a canonical federated learning benchmark, showing that these factorizations significantly improve the privacy/utility curve (in fact, closing 2/3rds of the gap to non-private training left by the previous state-of-the-art for single pass algorithms). For prefix sums, we show computationally-efficient structured matrices provide high-fidelity approximations to the optimal matrices, allowing implementations to scale essentially independent of the number of iterations $n$.

**Notation and conventions** Matrices will be denoted by bolded capital letters (e.g. $\mathbf{A}$, $\mathbf{B}$), with some symbols reserved for special matrices, notably $\mathbf{S}$ (prefix sums) and $\mathbf{M}$ (momentum, defined

in Section 4 and illustrated in Appendix A). Vectors will be denoted by bolded lowercase letters (e.g. $\mathbf{x}$, $\mathbf{y}$). For a real symmetric matrix $\mathbf{A}$, the smallest and the largest eigenvalues are denoted by $\lambda_{\min}(\mathbf{A})$ and $\lambda_{\max}(\mathbf{A})$. For a matrix $\mathbf{A}$, $\mathbf{A}^*$ denotes the conjugate transpose and $\mathbf{A}^\dagger$ denotes the Moore-Penrose pseudoinverse; a star as in $\mathbf{X}^\star$ indicates a matrix that is optimal in a way made clear by context. $\mathrm{diagpart}$ represents the operation of taking the diagonal elements of a matrix; $\mathrm{diag}$ represents embedding a vector argument on the diagonal of a matrix. Additional notation is summarized in Appendix A.

## 2 Privacy for adaptive streams

In this section, we provide structural results that clarify the classes of mechanisms/algorithms for cumulative sums and other linear computations in the *continual release model* [4, 3] that remain private even when the inputs stream is defined *adaptively*. In the continual release model, a mechanism receives a stream of inputs $\mathbf{G} = [\mathbf{g}_1, \ldots, \mathbf{g}_n]$ and produces a stream of outputs $\mathbf{a}_1, ..., \mathbf{a}_n$, where output $\mathbf{a}_i$ is intended to approximate some function of the prefix $\mathbf{g}_1, ..., \mathbf{g}_i$ and must be generated before $\mathbf{g}_{i+1}$ is received. We specify which parts of the input can depend on a single person's data via a *neighbor* relation $\mathcal{N}$ on data streams. Two streams are neighbors if they differ in one person's data. For example, if one person's data directly affects exactly one input in the stream ("*event-level privacy*"), then we say two data streams are *neighboring* if they differ in exactly one element (that is, they are at Hamming distance 1).

The original works on continual release analyzed privacy in a *nonadaptive* model: a mechanism $\mathcal{M}$ is $(\varepsilon, \delta)$-differentially private [31, 32] for nonadaptive continual release if, for all pairs of adjacent data streams $\mathbf{G}, \mathbf{H}$, the corresponding distributions on output streams $\mathcal{M}(\mathbf{G})$ and $\mathcal{M}(\mathbf{H})$ are $(\varepsilon, \delta)$-indistinguishable, denoted $\mathcal{M}(\mathbf{G}) \approx_{\varepsilon, \delta} \mathcal{M}(\mathbf{H})$. That is, for all events $E$, we have $\Pr[\mathcal{M}(\mathbf{G}) \in E] \leq e^\varepsilon \Pr[\mathcal{M}(\mathbf{H}) \in E] + \delta$ and $\Pr[\mathcal{M}(\mathbf{H}) \in E] \leq e^\varepsilon \Pr[\mathcal{M}(\mathbf{G}) \in E] + \delta$. (A variant of this definition tailored to differentially private gradient descent, with a precise instantiation of the neighborhood notion is presented in Definition J.1 in the appendix.)

In many use cases—including those arising in iterative gradient-based optimization algorithms—the nonadaptive model is inadequate, since the inputs $\mathbf{g}_i$ may be generated in real time as a function of previous outputs $\mathbf{a}_1, ..., \mathbf{a}_{i-1}$. To summarize the more general, adaptive definition [14, 8], consider an adversary that *adaptively* defines two input sequences $\mathbf{G} = (\mathbf{g}_1, \ldots, \mathbf{g}_n)$ and $\mathbf{H} = (\mathbf{h}_1, \ldots, \mathbf{h}_n)$. The adversary must satisfy the promise that these sequences correspond to neighboring data sets. The privacy game proceeds in rounds. At round $t$, the adversary generates $\mathbf{g}_t$ and $\mathbf{h}_t$. The game accepts these if the input streams defined so far are valid, meaning that there exist completions $(\tilde{\mathbf{g}}_{t+1}, ..., \tilde{\mathbf{g}}_n)$ and $(\tilde{\mathbf{h}}_{t+1}, ..., \tilde{\mathbf{h}}_n)$ so that $((\mathbf{g}_1, ..., \mathbf{g}_t, \tilde{\mathbf{g}}_{t+1}, ..., \tilde{\mathbf{g}}_n), (\mathbf{h}_1, ..., \mathbf{h}_t, \tilde{\mathbf{h}}_{t+1}, ..., \tilde{\mathbf{h}}_n)) \in \mathcal{N}$. For example, in the case of event-level privacy, the game simply checks that the two streams differ in at most one position so far.

The game is parameterized by a bit $\mathsf{side} \in \{0, 1\}$ which is unknown to the adversary but constant throughout the game. The game hands either $\mathbf{g}_t$ or $\mathbf{h}_t$ to the mechanism $\mathcal{M}$, depending on $\mathsf{side}$. The mechanism's output $\mathbf{a}_t$ is then sent to the adversary. The privacy requirement is that the adversary's views with $\mathsf{side} = 0$ and $\mathsf{side} = 1$ be $(\varepsilon, \delta)$ indistinguishable. One can substitute other relevant notions of indistinguishability like those from CDP [33, 34], Renyi DP [35], or Gaussian DP [36]. The mechanisms we consider generally satisfy Gaussian DP.

The nonadaptive version of the definition is weaker but easier to work with. It is therefore natural to look for classes of mechanisms for which the two definitions are equivalent (and thus for which a nonadaptive privacy proof implies the more general guarantee). We first observe that such a transfer statement holds for "pure" $\varepsilon$-DP (in which $\delta = 0$). We defer all the proofs to Appendix D.

**Proposition 2.1.** *Every mechanism that is $(\varepsilon, 0)$ nonadaptively DP in the continual release model satisfies the adaptive version of the definition, with the same parameters.*

Unfortunately, not all privacy proofs for the nonadaptive model transfer to the adaptive setting. Indeed, we show that there are additive-noise mechanisms (which simply add noise from a pre-defined distribution to some function of the data) that are nonadaptively $(\varepsilon, \delta)$-DP, but *not* private in the adaptive setting (Appendix D.1).

**Adaptive privacy for Gaussian noise addition mechanisms**   In Theorem 2.1 we show that every matrix mechanism with Gaussian noise addition that is $(\varepsilon, \delta)$-DP in the nonadaptive model is also private in the adaptive setting:

**Theorem 2.1.** *Let $\mathbf{A} \in \mathbb{R}^{n \times n}$ be a lower-triangular full-rank query matrix, and let $\mathbf{A} = \mathbf{BC}$ be any factorization with the following property: for any two neighboring streams of vectors $\mathbf{G}, \mathbf{H} \in \mathbb{R}^{n \times d}$, we have $\|\mathbf{C}(\mathbf{G} - \mathbf{H})\|_F \leq \kappa$. Let $\mathbf{Z} \sim \mathcal{N}(0, \kappa^2 \sigma^2)^{n \times d}$ with $\sigma$ large enough so that $\mathcal{M}(\mathbf{G}) = \mathbf{AG} + \mathbf{BZ} = \mathbf{B}(\mathbf{CG} + \mathbf{Z})$ satisfies $(\varepsilon, \delta)$-DP (or $\rho$-zCDP or $\mu$-Gaussian DP) in the nonadaptive continual release model. Then, $\mathcal{M}$ satisfies the same DP guarantee (with the same parameters) even when the rows of the input are chosen adaptively.*

To prove this we crucially use the rotational invariance of spherical Gaussian distribution, yielding distributional equivalence of an orbit of mechanisms: those factorizations expressible as $\mathbf{BUU}^*\mathbf{C}$ for $\mathbf{U}$ unitary. This observation can similarly be leveraged to show a subtly distinct fact:

**Proposition 2.2.** *For any factorization $\mathbf{A} = \mathbf{BC}$ where $\mathbf{A}$ is lower-triangular, there exists a factorization $\mathbf{A} = \widehat{\mathbf{B}}\widehat{\mathbf{C}}$ which induces a distributionally equivalent matrix mechanism under Gaussian noise with $\widehat{\mathbf{B}}$ and $\widehat{\mathbf{C}}$ lower triangular. This factorization can be explicitly computed from $\mathbf{A} = \mathbf{BC}$ via an appropriate LQ decomposition. Normalizing to all-nonnegative entries on the diagonal, this factorization is unique.*

## 3   Computing optimal factorizations

Once one writes down the matrix mechanism as in Eq. (1), it is natural to seek factorizations that minimize the error $\widehat{\mathbf{AG}} - \mathbf{AG}$ in some metric of choice. In this section we consider the expected squared reconstruction error of the estimate $\widehat{\mathbf{AG}}$, which has previously been noted as an appropriate formulation of error for the setting of training private ML models [6, Theorem 5]. Further, for simplicity we restrict our attention to the single-pass setting. That is, for the remainder of the paper we will assume:

**Definition 3.1.** *Two data matrices $\mathbf{G}$ and $\mathbf{H}$ in $\mathbb{R}^{n \times d}$ will be considered to be neighboring if they differ by a single row, with the $\ell_2$-norm of the difference in this row at most $\zeta$.*

Under this notion of sensitivity, one can make the estimation of any query $\mathbf{AG}$ $(\varepsilon, \delta)$-DP via Theorem 3.1, which via Theorem 2.1 immediately extends to adaptive streams. It is worth mentioning that while we state the analytic vairance for the Normal distribution to satisfy $(\varepsilon, \delta)$-DP, in practice we arrive at the required (and tighter) variance via empirical privacy accounting methods like zero concentrated DP (zCDP) accounting [33], or privacy loss distribution (PLD) accounting [37].

**Theorem 3.1** (Adapted from [1])**.** *Consider a query matrix $\mathbf{A} \in \mathbb{R}^{n \times n}$ along with a fixed factorization $\mathbf{A} = \mathbf{B}_{n \times n}\mathbf{C}_{n \times n}$ with $\gamma = \max_{i \in [n]} \left\|\mathbf{C}_{[:, i]}\right\|_2$, the maximum column norm of $\mathbf{C}$. Let $\mathbf{G} \in \mathbb{R}^{n \times d}$ be a fixed (non-adaptive) data matrix with each row of $\mathbf{G}$ having $\ell_2$-norm at most $\zeta$. The algorithm that outputs $\mathbf{B}(\mathbf{CG} + \mathbf{Z})$ with $\mathbf{Z} \sim \mathcal{N}\left(0, \frac{\gamma^2 \zeta^2 (2\log(1/\delta) + \varepsilon)}{\varepsilon^2}\right)^{n \times d}$ satisfies $(\varepsilon, \delta)$-DP.*

In this setting, with $\mathbf{Z} \sim \mathcal{N}\left(0, \gamma^2\right)^{n \times d}$ following Theorem 3.1 (which ensures a fixed level of privacy when $\zeta = 1$ for an arbitrary factorization factorization $\mathbf{A} = \mathbf{BC}$) the expected reconstruction error can be computed directly as $\mathcal{L}(\mathbf{B}, \mathbf{C})$ [1, Proposition 9], [27, Equation 3],

$$\gamma^2(\mathbf{C}) = \max_{i \in [1, \ldots, n]} \left\|\mathbf{C}_{[:, i]}\right\|_2^2 \qquad \text{and} \qquad \mathcal{L}(\mathbf{B}, \mathbf{C}) = \gamma^2(\mathbf{C}) \left\|\mathbf{B}\right\|_F^2. \qquad (2)$$

As has been noted [1, 27], Eq. (2) can be manipulated to yield a convex program, for which hand-tuned algorithms exist [27, Section 4]. Theorem 2.1 shows, for the first time, that arbitrary factorizations found by minimizing this optimization problem can be applied in the adaptive streaming setting.

We present an alternative characterization of these optima, which reformulates the optimization problem as a fixed-point problem. We show that simply iterating an explicit mapping converges to this fixed point from an appropriate initialization, and observe numerically that the associated algorithm achieves fast, global convergence.

Since the Moore-Penrose pseudoinverse yields the minimal $\ell_2$-norm solution to a set of underdetermined linear equations [38, Theorem 2.1.1], we note that for a fixed $\mathbf{C}$ term (of any dimensionality),

the optimal $\mathbf{B}$ may be expressed as $\mathbf{B}_{\mathbf{C}}^{\star} = \mathbf{A}\mathbf{C}^{\dagger}$. Since $\mathbf{A} = \mathbf{B}\mathbf{C}$ implies $\mathbf{A} = (\alpha\mathbf{B})(\frac{1}{\alpha}\mathbf{C})$, for any linear space of matrices $\mathbf{V}$, we may express the optimization problem of interest as

$$\min_{\mathbf{C}\in\mathbf{V}} \mathcal{L}\left(\mathbf{A}\mathbf{C}^{\dagger}, \mathbf{C}\right) = \min_{\mathbf{C}\in\mathbf{V}} \gamma^2(\mathbf{C})\left\|\mathbf{A}\mathbf{C}^{\dagger}\right\|_F^2 = \min_{\mathbf{C}\in\mathbf{V}, \gamma^2(\mathbf{C})=1}\left\|\mathbf{A}\mathbf{C}^{\dagger}\right\|_F^2. \tag{3}$$

The properties of the problem Eq. (3) have been studied previously. In particular, [27, Section 3] studied a symmetric version, transforming the problem as:

$$\mathbf{X}^{\star} = \underset{\mathbf{X} \text{ is PD}, \mathbf{X}_{[i,i]}\leq 1, 1\leq i\leq n}{\arg\min} \operatorname{tr}(\mathbf{A}^*\mathbf{A}\mathbf{X}^{-1}) \tag{4}$$

which essentially reparameterizes Eq. (3) (with $\mathbf{V} = \mathbb{R}^n$) in terms of $\mathbf{C}^*\mathbf{C}$. To recover a matrix-mechanism factorization of $\mathbf{A}$, then, one may utilize any $\mathbf{C}$ such that $\mathbf{X}^{\star} = \mathbf{C}^*\mathbf{C}$, e.g. $\mathbf{C} = \sqrt{\mathbf{X}^{\star}}$. Proposition 2.2 can be used to construct a lower-triangular factorization if desired.

Yuan et al. [27] show: 1) Any solution $\mathbf{X}^{\star}$ of Eq. (4) must have diagonal entries exactly 1. 2) Any solution $\mathbf{X}^{\star}$ may be taken to be strictly within the positive-definite cone, with minimal eigenvalue bounded from below in terms of the eigenvalues of $\mathbf{A}$. 3) For any full-rank $\mathbf{A}$, $\mathbf{X} \mapsto \operatorname{tr}(\mathbf{A}^*\mathbf{A}\mathbf{X}^{-1})$ is strictly convex over symmetric, positive-definite matrices. Therefore the solution to Eq. (4) is unique.

By analyzing Eq. (4) directly, we derive a characterization of solutions in terms of an explicit fixed-point problem, with a corresponding bound on the optimality gap.

**Theorem 3.2.** *The minimizer $\mathbf{X}^{\star}$ of Eq.* (4) *is in one-to-one correspondence with the unique fixed point of the function $\phi : \mathbb{R}_+^n \to \mathbb{R}_+^n$ defined by*

$$\phi(\mathbf{v}) = \operatorname{diagpart}\left(\sqrt{\operatorname{diag}(\mathbf{v})^{1/2}\,\mathbf{A}^*\mathbf{A}\,\operatorname{diag}(\mathbf{v})^{1/2}}\right). \tag{5}$$

*Letting*

$$\mathcal{X}(\mathbf{v}) = \operatorname{diag}(\mathbf{v})^{-1/2}\left(\operatorname{diag}(\mathbf{v})^{1/2}\,\mathbf{A}^*\mathbf{A}\,\operatorname{diag}(\mathbf{v})^{1/2}\right)^{1/2}\operatorname{diag}(\mathbf{v})^{-1/2}, \tag{6}$$

*for the fixed point $\mathbf{v}^{\star}$ of Eq.* (5)*, that is $\phi(\mathbf{v}^{\star}) = \mathbf{v}^{\star}$, we have $\mathbf{X}^{\star} = \mathcal{X}(\mathbf{v}^{\star})$, and this pair $(\mathbf{X}^{\star}, \mathbf{v}^{\star})$ satisfies*

$$\mathbf{A}^*\mathbf{A} = \mathbf{X}^{\star}\operatorname{diag}(\mathbf{v}^{\star})\mathbf{X}^{\star}. \tag{7}$$

*Further, for any $\mathbf{v} \in \mathbb{R}_+^n$, the objective value of the primal problem Eq.* (4) *is lower-bounded by*

$$\operatorname{tr}\left(\operatorname{diag}(\mathbf{v})(2\mathcal{X}(\mathbf{v}) - \mathbf{I})\right), \tag{8}$$

*and this bound is tight for $\mathbf{v} = \mathbf{v}^{\star}$.*

The sum of the elements of $\phi$ represents a quantity of independent interest in quantum information, the so-called Jozsa fidelity [39, 40], while $\mathcal{X}$ represents the matrix geometric mean of $\mathbf{A}^*\mathbf{A}$ and $\operatorname{diag}(\mathbf{v})^{-1}$ [41, 42]. These connections give some hope that the fixed point of $\phi$ can be understood in a direct manner. We can show local, though not yet global, convergence of the iterates of $\phi$ to this fixed point.

**Theorem 3.3.** *$\phi$ defined in Eq.* (5) *is a local contraction around its fixed point in a suitable metric, and hence there is a neighborhood of this fixed point in which iterates of $\phi$ converge to this fixed point. The precise norm of this contraction, and the size of the neighborhood in which convergence is guaranteed, can both be estimated in terms of minimum and maximum eigenvalues of $\mathbf{A}$.*

This result can be shown by linearizing the mapping $\phi$ around its fixed point and performing an involved estimate of its Jacobian at the fixed point. As such, it is implied by Theorem E.1, which states that the linearization of $\phi$ around its fixed point is a contraction in a suitable metric, and therefore the Banach fixed-point theorem applies.

**Remark.** Notably missing from quantification of the contraction is the dimension $n$. Indeed, the argument is dimension-independent in a strong sense: it applies to the suitably generalized definition of $\phi$ where $\mathbf{A}$ is any bounded linear operator on a Hilbert space with bounded inverse.

**Algorithm 1** DP Matrix Factorization SGD

1: Inputs:
2:     factorization $\mathbf{M} = \mathbf{BC}$
3:     overall learning rate $\eta$
4:     noise level $\sigma$, clipping norm $\zeta$
5:     examples $\chi_i$, $i \in \{1, \ldots, n\}$
6: $\boldsymbol{\theta}_{[0,:]} := 0 \in \mathbb{R}^d$
7: Sample $\mathbf{Z} \in \mathbb{R}^{n \times d}$, $\mathbf{Z}_{[i,j]} \sim \mathcal{N}(0, \sigma^2)$ iid
8: **for** $i$ in $1, \ldots, n$ **do**
9:     $\hat{\mathbf{g}} := \nabla_{\boldsymbol{\theta}_{[i-1,:]}} \ell(\boldsymbol{\theta}; \chi_i)$
10:     $\mathbf{G}_{[i,:]} := \hat{\mathbf{g}} \cdot \min\left\{\frac{\zeta}{\|\hat{\mathbf{g}}\|_2}, 1\right\}$
11:     $\boldsymbol{\theta}_{[i,:]} := -\eta\left(\mathbf{M}_{[i,:]} \mathbf{G}_{[1:i,:]} + \mathbf{B}_{[i,:]} \mathbf{Z}\right)$

**Algorithm 2** Heavy-ball momentum

1: $\boldsymbol{\theta}_0 := 0$, $\mathbf{m}_0 := 0$
2: **for** $i$ in $1, \ldots, n$ **do**
3:     $\mathbf{m}_i := \beta \cdot \mathbf{m}_{i-1} + \mathbf{g}_i$
4:     $\boldsymbol{\theta}_i := \boldsymbol{\theta}_{i-1} - \eta_i \mathbf{m}_i$

**Empirical performance of the fixed-point method** Experimentally, iterating the mapping $\phi$ is sufficient to converge to the global optimum extremely quickly from any initial point (modulo potential numerical issues in computing the matrix square root, discussed in Appendix E.4). Yuan et al. [27, Algorithm 1] design an algorithm with globally linear and locally quadratic convergence rate, with similar asymptotics to iterating $\phi$ (each dominated by an $n^3$ term), though at the cost of introducing a parameter $T$. Iterating $\phi$, on the other hand, is parameter-free. We implemented [27, Algorithm 1] as well as a direct gradient-descent-based method to numerically compare convergence speed. As a canonical benchmark, we computed optimal factorizations of the $512 \times 512$ and $2048 \times 2048$ prefix-sum matrices $\mathbf{S}$ (Appendix G provides a visualization of this optimal factorization). Our fixed-point algorithm was significantly faster than either of the alternatives to compute optima, computing a lower-loss matrix for the larger problem in less than 3 minutes than either alternative found in over 80 minutes. See Appendix E.4 for details. Further, via Eq. (8), our approach provides an optimality certificate that allows a precise specification of the stopping criteria in terms of any target optimality gap. The speed and simplicity of our fixed-point algorithm was a significant enabler of the mechanism exploration presented in the next section.

## 4 The matrix mechanism for SGD

To define our gradient descent algorithm, let $\mathbf{G} \in \mathbb{R}^{n \times d}$ be the matrix of gradients, with row vector $\mathbf{g}_i \in \mathbb{R}^{1 \times d}$ the gradient observed on iteration $i$ after clipping to norm at most $\zeta$; we abuse notation slightly by writing $\mathbf{G}_{[1:i,:]} \in \mathbb{R}^{n \times d}$, formed by taking the first $i$ rows of $\mathbf{G}$, with zeros for the as-of-yet unobserved gradient rows for iterations $i + 1, \ldots, n$ (the lower triangular structure of the matrices we consider will imply $\mathbf{G}_{[1:i,:]}$ vs $\mathbf{G}$ does not in fact change the value computed). With this notation, we define Algorithm 1, a general template for private SGD algorithms. The power of this general formulation comes largely from the following privacy guarantee:

**Theorem 4.1.** *Under the "replace with zero" notion of differential privacy (in Defininition J.1 in the appendix) over examples (records) $\chi_i$, taking $\hat{\mathbf{g}} = 0$ when $\chi_i = \bot$, Algorithm 1 (that releases the iterates $\boldsymbol{\theta}_{[i,:]}$) satisfies equivalent $(\varepsilon, \delta)$-DP to the Gaussian mechanism with noise variance $\sigma^2$ applied to records with $\ell_2$ sensitivity at most $\zeta\gamma$, where $\gamma = \max_i \left\|\mathbf{C}_{[:,i]}\right\|_2$ is the maximum column norm of $\mathbf{C}$, and $\zeta$ is the clipping norm.*

Theorem 4.1 shows that the contribution of $\mathbf{C}$ to the loss Eq. (19) is reflected in the privacy guarantee of Algorithm 1, by determining the sensitivity of the matrix mechanism Eq. (1). The contribution of $\mathbf{B}$ determines the expected squared reconstruction error of the matrix mechanism, by definition. This quantification can in turn be converted by existing analytical methods to a regret bound for convex losses:

**Proposition 4.1** (Adaptation of Theorem C.1, Kairouz et al. [6]). *In the setup of Algorithm 1, let $\mathbf{M}$ be the prefix-sum matrix $\mathbf{S}$, and assume $\ell$ is convex with $\ell_2$-Lipschitz constant $L$. Let $\theta_t = \boldsymbol{\theta}_{[t,:]}$. For any $\theta^\star \in \mathbb{R}^d$,*

| Mechanism | $\mathbf{B}$ | $\mathbf{C}$ | s.t. $\mathbf{M} = \mathbf{BC}$ |
|---|---|---|---|
| `Honaker Online` | $\mathbf{MS}^{-1}\mathbf{B}_{\text{hs}}$ | $\mathbf{C}_{\mathcal{T}}$ | Equivalent to DP-FTRL of Kairouz et al. [6] |
| `Honaker Full` | $\mathbf{MS}^{-1}\mathbf{B}_{\text{hf}}$ | $\mathbf{C}_{\mathcal{T}}$ | |
| `Opt Prefix Sum` | $\mathbf{MS}^{-1}\mathbf{B}_S^*$ | $\mathbf{C}_S^\star$ | for optimal $\mathbf{S} = \mathbf{B}_S^\star \mathbf{C}_S^\star$ |
| `Optimal M = B C` | $\mathbf{B}_M^\star$ | $\mathbf{C}_M^\star$ | for optimal $\mathbf{M} = \mathbf{B}_M^\star \mathbf{C}_M^\star$ |

Table 1: Instantiations of Algorithm 1 for various factorizations of the SGD matrix $\mathbf{M} = \mathbf{BC}$.

$$\frac{1}{n}\sum_{t=1}^{n} \mathbb{E}\left[\ell(\theta_t; \chi_t) - \ell(\theta^\star; \chi_t)\right] \leq \eta L^2 + \frac{1}{2\eta n}\left(\|\theta^\star\|_2^2 - \|\theta_1\|_2^2\right) + \frac{L\sigma\eta}{\sqrt{n}}\|\mathbf{B}\|_F$$

We now consider different instantiations of Algorithm 1. Let $\mathbf{S}$ be the prefix-sum matrix as defined in Appendix A, and let $\mathbf{C}_{\mathcal{T}}$ be the matrix representation of the binary tree (Appendix C), so for appropriate choices of the reconstruction matrices $\mathbf{B}_{\text{hs}}$ and $\mathbf{B}_{\text{hf}}$, $\mathbf{S} = \mathbf{B}_{\text{hs}}\mathbf{C}_{\mathcal{T}}$ gives the `Honaker Online` mehcanism, and $\mathbf{S} = \mathbf{B}_{\text{hf}}\mathbf{C}_{\mathcal{T}}$ gives the `Honaker Full` mechanism. In particular, using the `Honaker Online` factorization in Algorithm 1 recovers the non-momentum DP-FTRL algorithm of Kairouz et al. [6].

However, Kairouz et al. [6] observed that for non-convex objectives, DP-FTRL with momentum provided superior privacy/accuracy tradeoffs. Given prefix sums, momentum can be implemented as post processing by estimating individual gradients/updates as the difference of successive cumulative sums (multiplication by $\mathbf{S}^{-1}$), and then passing these into a standard momentum SGD optimizer. We show that performance can be improved by directly incorporating momentum and a-priori learning rate schedules directly into the DP mechanism.

A basic but important observation is that momentum SGD can be expressed as a linear map of gradients $\mathbf{G} \to \mathbf{MG}$. We consider the classic momentum algorithm of Polyak [43], with per-iteration learning rates[3] $\eta_1, \ldots, \eta_n$ and momentum $\beta \in [0, 1)$, as in Algorithm 2. Alternatively, we can express momentum SGD as a linear operator on the gradients:

**Proposition 4.2.** *For any $\beta \in [0, 1)$, $n \geq 1$, per iteration learning rates $\eta_1, \ldots, \eta_n$, define the lower-triangular matrix $\mathbf{M} \in \mathbb{R}^{n \times n}$ as the product of lower-triangular matrices $\mathbf{M}^{(\eta)}$ and $\mathbf{M}^{(\beta)}$:*

$$\mathbf{M}^{(\eta)}_{[i,j]} = \begin{cases} \eta_j & i \geq j \\ 0 & otherwise \end{cases}, \quad \mathbf{M}^{(\beta)}_{[i,j]} = \begin{cases} \beta^{i-j} & i \geq j \\ 0 & otherwise \end{cases}, \quad and \quad \mathbf{M} = \mathbf{M}^{(\eta)}\mathbf{M}^{(\beta)}. \quad (9)$$

*Then, for any matrix of of per-iteration gradients $\mathbf{G} \in \mathbb{R}^{n \times d}$ with rows $[\mathbf{g}_1, \ldots, \mathbf{g}_n]$, the sequence of iterates $\boldsymbol{\theta} \in \mathbb{R}^{n \times d}$ with rows $[\boldsymbol{\theta}_1, \ldots \boldsymbol{\theta}_n]$ produced by Algorithm 2 can equivalently be written $\boldsymbol{\theta} = -\mathbf{MG}$.*

We apply the fixed-point algorithm of Section 3 to $\mathbf{M}$ to obtain optimal matrix mechanisms $\mathbf{M} = \mathbf{BC}$, which indeed leads to improved performance.[4] We can also convert mechanisms that produce DP prefix sums to produce momentum iterates via post processing: given any matrix mechanism for the prefix sum problem given as a factorization $\mathbf{S} = \mathbf{BC}$, we can convert this to a mechanism for momentum as $\mathbf{M} = \hat{\mathbf{B}}\mathbf{C}$ where $\hat{\mathbf{B}} = \mathbf{MS}^{-1}\mathbf{B}$. Straightforward calculations show this representation is equivalent to the "Momentum Variant" of DP-FTRL [6]. This allows us to consistently evaluate the mechanisms in terms of the total variance (squared error) induced in the outputs by unit-variance noise $\mathbf{Z}$ in the mechanism via Eq. (2). Table 1 summarizes the four instantiations of Algorithm 1 for any choise of $\mathbf{M}$; note in particular that for $\beta = 0$ and a fixed learning rate schedule $\eta_i = 1$, $\mathbf{M} = \mathbf{S}$. Fig. 1 compares the per-step squared error of the DP momentum iterates for these methods for $\beta = 0$ (prefix sums, as a constant $\eta = 1$ is used) and $\beta = 0.95$.

**Computational efficiency** Some care is necessary for the efficient implementation of Algorithm 1, particularly the computation of $\mathbf{M}_{[i,:]}\mathbf{G}_{[1:i,j]} + \mathbf{B}_{[i,:]}\mathbf{Z}$ on line 11. First, we observe that via Proposition 4.2 we can efficiently compute the $\mathbf{MG}$ term via Algorithm 2, rather than as a matrix operation.

---

[3]The schedule may be arbitrary, but must be chosen a priori in a data-independent way.
[4]The definition $\mathbf{M} = \mathbf{M}^{(\eta)}\mathbf{M}^{(\beta)}$ is for the convenience, and is unrelated to the optimal factorization.

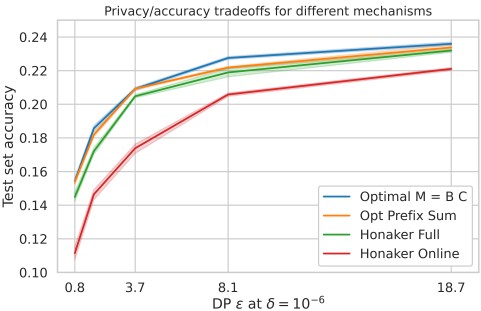
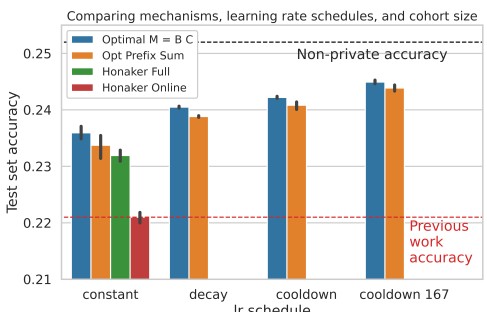

Figure 2: Test accuracy for the StackOverflow next-word-prediction task. A grid search over client and server learning rates and momentum $\beta$, with the best hyperparameters selected based on validation set accuracy. We then re-ran 11 repetitions with the best hyperparameters and report the mean test set accuracy with confidence intervals. All models trained with 100 clients per round and a constant learning rate schedule.

Figure 3: Test accuracy for mechanisms incorporating learning rate decay in the manner of Section 4 at $\varepsilon = 18.9$. The red horizontal line represents test accuracy of previous state of the art at $\varepsilon = 18.9$; highest horizontal line, test accuracy of non-private model. The final bar group shows learning rate cooldown with 167 clients/round, the maximum possible for a single training pass of 2048 training rounds.

This leaves the computation of the noise $\mathbf{B}_{[i,:]}\mathbf{Z}$. For applications to ML, $d$ could be $10^6 - 10^9$, and with even $n = 10^4$ rounds (iterations) this might make the total calculation quite expensive if not prohibitive. With an efficient TensorFlow implementation, in our experiments with $d \approx 4 \times 10^6$ and $n = 2048$, we found we could compute the noise directly. However, for larger applications we show (in Appendix H) that one can compute structured matrices that well-approximate the optimal $\mathbf{B}^\star$ for the prefix sum matrix while allowing for $\mathcal{O}(d)$ calculation of the per-round noise vectors (on par with that of the binary tree mechanism, which can also provide computational efficiency with a careful implementation, see Table 2). The key is to observe the diagonal dominance of the optimal $\mathbf{B}^\star$ (see Appendix G), leading to an approximation $\hat{\mathbf{B}}$ that is the sum of a lower-triangular $d$-banded matrix with the remaining entries in the lower triangle extracted from a low-rank approximation computed via alternating-least-squares.

## 5 Experimental results

The results of Sections 2 and 4 significantly expand the space of mechanisms which can be used in training ML models with differential privacy in the single-pass setting. In this section we demonstrate that these techniques can in fact significantly advance the state-of-the-art in private ML.

**User-level privacy for language models** Private training is particularly important for generative language models: training language models on data from the right distribution is critical for utility (e.g., user input in a mobile keyboard [44, 45]), but this data is often privacy-sensitive. Further, language models have been shown to be capable of memorizing training data [46–48]. In this setting it is important to consider user-level DP, where the neighbor relation of the DP guarantee covers all of the training examples (tokens) from any one user, as opposed to a single training example [49]. In our setting, this corresponds to ensuring that each user's examples contribute to a bounded $\ell_2$-norm update to a single row of $\mathbf{X}$ (Definition 3.1). This is accomplished by extending Algorithm 1 in the natural way to Federated Averaging [50]: instead of a single gradient, we take $\mathbf{g}_i$ to be the sum of the individually-clipped-to-$\zeta$ updates of all users (100 or 167 in our experiments) participating in the current round, with each user contributing to a single round over the course of training.

For these reasons, we focused on the StackOverflow next-word prediction problem, introduced in [51] and publicly hosted in TensorFlow-Federated (TFF) [52]. This task was explored extensively in [6], and serves as a major benchmark in federated learning, used in [51, 6, 53–55], and others. The StackOverflow dataset contains sufficiently many clients to support single-pass algorithms with 100

clients per round, similar to the baseline setup of [6]. For this reason, we are able to provide true $(\varepsilon, \delta)$ privacy quantifications for the models we train.[5]

**Results** We compare the four mechanisms of Table 1 on this problem; full experimental methodology, as well as additional plots (e.g., for validation error vs. training rounds) are provided in Appendix I. Fig. 2 shows that even with constant learning rates, our matrix factorization approaches significantly outperform the previous state-of-the-art across a range of privacy $\varepsilon$'s. Further, thanks to the results of Section 2, we are able to apply the `Honaker Full` mechanism for comparison. Fig. 3 shows that applying learning rate decay (dropping the learning rate by $0.15\times$ for the last 512 rounds) and learning rate cooldown (linearly dropping the learning rate from $1.0\times$ to $0.05\times$ over the last 512 rounds) show added improvements. These experiments all used 100 clients/round as in [6], but the rightmost bars shows increasing this to 167 (the maximum possible for a single pass of 2048 rounds) provides additional accuracy. In combination, these techniques close more than 2/3rds of the gap between private and non-private training.

## 6  Conclusions

We have shown the general applicability of the Gaussian matrix mechanism to the adaptive streaming setting, introduced a highly efficient mechanism of determining optimal (in the sense of total $\ell_2^2$ error) matrix mechanisms, used this approach to directly incorporate momentum and learning rate schedules into the DP mechanism, and empirically demonstrated the resulting private SGD (or FedAvg in the federated setting) substantially improves on the state of the art for private ML.

While our focus has been on the application of these techniques to gradient-based optimization algorithms, we emphasize that the problem of producing private estimates for linear queries in the adaptive streaming setting is a fundamental DP primitive of much broader applicability, as noted in the introduction, and so our work immediately leads to improvements in those applications as well.

Finally, our work raises numerous natural follow-up questions which we hope will inspire subsequent work; we sketch these in Appendix B.

## Acknowledgements

We thank Zachary Charles, Thomas Steinke, Jonathan Ullman, Zheng Xu, and Anastasia Koloskova for their valuable feedback and insights. In particular Thomas pointed us to the Speyer's argument of the lower bound; Zach discussed several of the technical arguments with the authors; Jon provided helpful insights into alternate proofs of Theorem 2.1; Zheng provided valuable pointers to code which we were able to leverage; and Anastasia caught a significantly dropped term in Proposition 4.1.

Adam Smith was supported in part by NSF award CNS-2120667 and gifts from Apple and Google.

Sergey Denisov was supported by NSF award DMS-2054465 and Van Vleck Professorship research award.

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
