# OpenReview forum: "Improved Differential Privacy for SGD via Optimal Private Linear Operators on Adaptive Streams"
_NeurIPS.cc/2022/Conference — NeurIPS 2022 Accept_

### Official Review · Reviewer_9W6b · 2022-07-11

**Rating:** 7
**Confidence:** 3
**Soundness:** 3 good
**Presentation:** 4 excellent
**Contribution:** 4 excellent

**Summary:**

This work studies matrix mechanism which guarantees differential privacy over adaptive streams. The authors prove that Gaussian mechanism and $(\epsilon, 0)$-DP mechanisms that are private for non-adaptive inputs are also private for adaptive inputs. The authors further characterize the expression for optimal matrix factorization. Based on the theoretical intuitions, the authors design a matrix mechanism for SGD which has superior performance than previous algorithms.

**Questions:**

Questions
- For the fixed-point algorithm, what's the rough convergence rate, and what's the main difficulty in proving a formal convergence guarantee?
- How would a sub-optimal matrix factorization affect algorithm performance? For example, if $\phi$ is some distance $\Delta$ away from the true optimum, is it possible to characterize the performance degredation?
- Is it possible to prove convergence for SGD with matrix factorization on say, convex and strongly convex loss functions?

Typos
- Second to last line in Figure 1 caption: mechaism -> mechanism

**Limitations:**

 Limitations and potential negative societal impact are adequately addressed.

**Strengths And Weaknesses:**

Strengths
- Designing private algorithms for adaptive streaming is an important problem in differential privacy. The authors prove fundamental theoretical results for the matrix mechanism that lead to a practical algorithm for SGD which empirically outperform existing methods.
- The paper is well structured and clearly written.

Weakness
- Some algorithms do not have formal theoretical guarantee.
  - Convergence guarantee for the fixed-point optimization algorithm used in experiments.
  - Effect of a sub-optimal matrix factorization on algorithm performance.
  - Convergence rate for DP Factorization for SGD (Algorithm 1)

---

> ### Author Response · Authors · 2022-08-02
> **We thank the reviewer for their reading, and answer their questions.**
>
> 1. Q1: For the fixed-point algorithm, what's the rough convergence rate, and what's the main difficulty in proving a formal convergence guarantee?
>
> *  Locally, the Banach fixed-point theorem gives a linear convergence rate, though it is possible that this analysis is missing something, since this fixed-point method empirically outperforms the Newton-step based method.
>
> * From the fixed-point perspective, the main difficulty in showing global convergence is the diagpart element of the mapping $\phi$. In some sense, the operation of this diagpart is dependent on the relationship between the geometry of the eigenbasis for $A^\top A$ and the standard basis. For example, if $A$ is diagonal, iterating the fixed-point mapping reduces to iterating (a scaled version of) the square root mapping element-by-element, for which it is easy to show global convergence (and in fact explicitly compute the fixed point). Whether such a simultaneous-diagonalization approach can be used in general is an interesting question for future work.
>
> 2. Q2: How would a sub-optimal matrix factorization affect algorithm performance? For example, if is some distance  away from the true optimum, is it possible to characterize the performance degradation?
>
> * We apologize for the lack of explicit quantifications for this question in our current draft. The way that suboptimal factorizations can be related to gradient-descent performance was additionally the subject of the first two questions from reviewer r9bX; we believe that the answers to those questions cover the questions of reviewer 9W6b as well. We will update our draft to make clear the relationships between the optimized loss $\mathcal{L}$, defined in equation (2), and known analyses of noisy gradient descent.
>
> 3. Q3: Is it possible to prove convergence for SGD with matrix factorization on say, convex and strongly convex loss functions?
>
> * It is. This can generally be inherited from the mapping between DP-FTRL and DP-SGD discussed in section 4.2 of [Kairouz et al. 2021](https://arxiv.org/abs/2103.00039). A more detailed discussion of the relationship between our matrix-factorization based approach, known DP-FTRL analyses, and DP-SGD can be found in our answers to reviewer r9bX’s first and second questions. Generally speaking, improvements in the factorization loss can be converted to improvements in bounds for the associated optimization problems. We will ensure to update our draft to contain clear statements and discussion of this relationship.

---

### Official Review · Reviewer_fTea · 2022-07-11

**Rating:** 6
**Confidence:** 2
**Soundness:** 3 good
**Presentation:** 2 fair
**Contribution:** 4 excellent

**Summary:**

The work analyzes the matrix mechanism in the adaptive streaming setting that Gaussian noise addition mechanism can be private in the adaptive streaming setting and designs a parameter-free ﬁxed-point algorithm to compute optimal factorizations for summation-style problems. Furthermore, the work provides user-level privacy experiment to see the improvement via the approach.


**Questions:**

I have no questions on the paper

**Limitations:**

The paper is well writing but not easy to understand, since the structure of it is not very clear.


**Strengths And Weaknesses:**

1. Adaptive privacy analysis is essential and inevitable while most works on continual release analyzed privacy only in a nonadaptive setting.
2. Theoretical results are solid to demonstrate efficiency and convergence of the proposed algorithm on the matrix mechanism, and DP in the adaptive streaming settings.

---

> ### Author Response · Authors · 2022-08-02
> **We thank the reviewer for insightful comments.**
>
> We will be happy to answer any questions that arise during the discussion.

---

### Official Review · Reviewer_r9bX · 2022-07-13

**Rating:** 6
**Confidence:** 3
**Soundness:** 3 good
**Presentation:** 3 good
**Contribution:** 3 good

**Summary:**

This paper studies the problem of private data analysis in the adaptive/streaming setting. More specifically, the authors provide a matrix factorization based method in this setting and apply it to SGD. The proposed method seems to be novel, and the empirical results look promising.

**Questions:**

The proposed method is novel, and I have the following concerns about the current paper:
1. How the different factorizations will affect the privacy and utility trade-off?
2. How the different factorizations will affect the convergence of SGD?
3. Whether the method can be extended to the case with multiple data passes?
4. In the experiments, how do you apply your method to FedAvg? Do you run several local steps and communicate the parameters, or do you just communicate the gradients? Do you only use one pass of the data? What is the number of communications of different methods?

**Limitations:**

Yes

**Strengths And Weaknesses:**

Strengths:
1. The proposed matrix factorization based method seems novel and can recover many existing methods for SGD.
2. The authors provide an efficient algorithm to estimate the optimal factorization.

Weaknesses:
1. It is unclear how the general factorization will affect the privacy and utility trade-off when applying it to SGD.
2. The method seems to be limited in the case that only one data pass.
3. The extension of the current method to the FedAvg in the experiments is unclear.

---

> ### Author Response · Authors · 2022-08-02
> **We thank the reviewer for their reading, and answer their questions.**
>
> 1. Q1: How the different factorizations will affect the privacy and utility trade-off?
>
> * We acknowledge that the relationship discussed below is generally missing from our current draft; we will update with a new version containing a clear discussion of the way the loss improvements discussed in our paper can be translated into improved understanding of the privacy/utility trade-off.
>
> * The loss $\mathcal{L}$ defined in equation (2) is intended to capture this dependence. Precisely, for a fixed level of privacy, and a fixed linear operator $A$, the loss $\mathcal{L}$ evaluated on a factorization $A = BC$ represents the expected squared reconstruction error of the associated matrix mechanism.
>
> * The particular choice of loss $\mathcal{L}$ is motivated from the analysis of DP-FTRL in [Kairouz et al. 2021](https://arxiv.org/abs/2103.00039). In particular, the analysis of Theorem 5.1 in [Kairouz et al. 2021](https://arxiv.org/abs/2103.00039) distills the error in the excess population risk through the average of the $\ell_2$-norm of noise vectors $b_t$ added at each time step $t$, which can be (potentially lossily) converted to the loss $\mathcal{L}$ we optimize; using Jensen’s inequality, this average is $\leq \sqrt{\frac{1}{T} \sum_{t=1}^T \|b_t\|_2^2}$. Via this conversion, improvements in the loss $\mathcal{L}$ we use, can be converted into improvements in the upper bound for convergence of DP FTRL. The empirical evaluations in section 5 line up well with this conversion.
>
> 2. Q2: How the different factorizations will affect the convergence of SGD?
>
> * The analysis for DP-FTRL mentioned above additionally maps between DP-FTRL and noisy SGD, in section 4.2 and appendix B.3. Using this mapping we can show that the the asymptotic rate of convergence remains the same as non-private SGD, but the steady state error to which DP-FTRL reaches depends on the specific matrix factorization approach (better factorizations having lower steady-state error). We will add a formal comment about this in the next iteration of the paper.
>
> 3. Q3: Whether the method can be extended to the case with multiple data passes?
>
> * This is an excellent question. There are at least three ways that the algorithms and methods here can be extended to handle multiple passes, which we discuss below. However, to keep the current paper focused on designing continual observation algorithms in the adaptive DP setting, we keep the question on optimally handling multiple participations as a question for future exploration. First, using standard composition/$\ell_2$-sensitivity based accounting for Gaussian mechanism. Second, estimating the worst case sensitivity for some fixed matrix $C$ (based on the factorization $A=BC$), and all possible user participation patterns.  A simplified version of this approach was followed in analyzing the binary-tree based mechanism used in DP-FTRL under the multi-epoch setting (Appendix D of [Kairouz et al. 2021](https://arxiv.org/abs/2103.00039)). Third, incorporating the user participation patterns to alter the original loss function $\mathcal{L}$ itself, to obtain better factorizations of the matrix $A$.
>
> 4. Q4: In the experiments, how do you apply your method to FedAvg? Do you run several local steps and communicate the parameters, or do you just communicate the gradients? Do you only use one pass of the data? What is the number of communications of different methods?
>
> * Apologies, this discussion is somewhat buried. At the end of the final full paragraph on page 8 (under the heading "User-level privacy for language models''), we include a description of the mapping to FedAvg. In response to the specific questions:
>   * We run several local steps and communicate the parameter updates, not the gradients. Precise settings are embedded in the code; we ran for 1 local epoch, with a max of 256 sentences per user, and a training batch size of 16 (so at most 16 steps on each client in a given round).
>   * The number of clients per round was carefully chosen to ensure a *true* single-pass training algorithm for StackOverflow; this is the reason we did not go over 167 clients / round. This guarantee is what allowed us to produce true epsilon-delta numbers while comparing algorithms.
>   * Each of the methods ran for the same number of communication rounds, 2048 (and have identical communication-resource requirements).

---

### Meta-Review · Area_Chair_dVz7 · 2022-08-22

**Recommendation:** Accept
**Confidence:** Certain

**Metareview:**

The paper applies the idea of matrix mechanism to the problem of DP learning which releases sequences of adaptively chosen gradients. This can be viewed as a generalization (and refinement) of the DP-FTRL approach. The approach is interesting and the empirical results are stronger than existing alternatives such as DP-SGD and DP-FTRL.

The idea is similar to a recent work [FHU22] https://arxiv.org/abs/2202.11205 on the problem of continual release of counting queries, but the authors have claimed independence and carefully explained the differences.

All reviewers are supporting acceptance and so do I.

**Award:**

No

---

### Decision · Program_Chairs · 2022-09-14

Accept